# Parent’s Health Locus of Control and Its Association with Parents and Infants Characteristics: An Observational Study

**DOI:** 10.3390/ijerph19105804

**Published:** 2022-05-10

**Authors:** Daniela Morniroli, Patrizio Sannino, Serena Rampini, Elena Nicoletta Bezze, Eleonora Milotta, Silvia Poggetti, Paola Marchisio, Samantha Bosis, Laura Plevani, Fabio Mosca, Maria Lorella Giannì

**Affiliations:** 1Department of Clinical Sciences and Community Health, Università degli Studi di Milano, 20122 Milan, Italy; milotta.eleonora14@gmail.com (E.M.); silviapoggetti@yahoo.it (S.P.); fabio.mosca@unimi.it (F.M.); maria.gianni@unimi.it (M.L.G.); 2Fondazione IRCCS Ca’ Granda Ospedale Maggiore Policlinico, NICU, 20122 Milan, Italy; elena.bezze@policlinico.mi.it (E.N.B.); laura.plevani@policlinico.mi.it (L.P.); 3Direzione Professioni Sanitarie, Fondazione IRCCS Ca’ Granda Ospedale Maggiore Policlinico, 20122 Milan, Italy; patrizio.sannino@unimi.it (P.S.); serena.rampini@unimi.it (S.R.); 4Fondazione IRCCS Ca’ Granda Ospedale Maggiore Policlinico, Pediatric Highly Intensive Care Unit, 20122 Milan, Italy; paola.marchisio@unimi.it (P.M.); samantha.bosis@policlinico.mi.it (S.B.); 5Department of Pathophysiology and Transplantation, University of Milan, 20122 Milan, Italy

**Keywords:** parent’s choices, health locus of control, preventive health behaviors, internal-external locus, parenting

## Abstract

The Parent Health Locus of Control (PHLOC) investigates the individual’s beliefs about the factors that govern their state of health and that of their children. The direct association between PHLOC and preventive health behaviours compliance has already been demonstrated in the literature. However, it is still unclear how socio-demographic variables affect the PHLOC. We investigated the Parent Health Locus of Control of parents of full-term and preterm infants and evaluated whether there were any correlations between PHLOC and socio-demographic characteristics of both parents and infants. A single-centre transverse observational study was conducted in the Neonatology Operating Unit IRCCS Ca ‘Granda Foundation Ospedale Maggiore Policlinico of Milan. A self-administered questionnaire of the PHLOC scale was distributed to a sample of 370 parents of 320 full-term and 52 preterm infants attending the follow-up service. Parents under the age of 36 and with a higher level of education (bachelor’s degree or above) believe less in the influence of the media on their child’s health. Parents of preterm and first-child infants recognize the greater influence of health care workers, while parents of newborns that have experience complications in their clinical course, believe more in the influence of fate (Chance Health Locus of Control) and God. Younger parents with a higher level of education may be more prone to healthy preventative behaviours. Preterm birth is positively associated with an increased trust in health care professionals. The experience of disease can increase a “Chance Health Locus of Control” and risky behaviours. Assessment of PHLOC helps identify categories of parents prone to risky health behaviours and offer targeted health education interventions.

## 1. Introduction

The maintenance of a newborn’s state of health is entrusted to the parents, whose actions have a significant weight on the psychophysical development of the infant. In particular, the “first 1000 days of life” represent a critical period, in the double meaning of a window of vulnerability, but also of opportunities to take specific actions to favour the psychophysical health potential of children [1]. Parents provide direct care, are responsible for accessing health services and, with their behaviours and lifestyles, represent a reference model for their children which can influence the infants’ future health choices. For example, vaccine hesitancy is a complex and context-specific phenomenon, which can also affect psychological factors, such as parents’ perception about the elements that govern their child’s health [2,3].

In recent years, the concept of ‘Locus of Control’ has been recognized as a psychological aspect with important implications for health. This concept originates from the psychological literature, particularly from the theory of social learning developed by Rotter in 1966, which describes the Locus of Control as individual beliefs about the control of life events [4]. Individuals who feel personally responsible for the events that happen to them are called internal; on the contrary, those who believe that the results they obtain are determined by forces beyond their will (external people, God, fate) are defined external [4,5]. The acquisition of a sense of control over life events is part of the personal growth process and is influenced by the surrounding environment, the family and social context in which one lives and the education one receives [6].

The concept of control place was also applied to the health context, with the definition of the health locus of control as a specific control construct that individual attributes to the various factors that govern his state of health [7]. A person may believe that his well-being depends on his choices and behaviours and that, therefore, the factors that affect his state of health are internal to him and under his control. On the contrary, others may think that external and uncontrollable elements, such as luck, destiny, God and other people, such as health workers and family members, can affect their well-being [2,6,7]. In the first case, we speak of internal HLOC, as health is considered dependent on factors internal to the individual. In the second case, we speak of external HLOC (External HLOC) as the state of health is considered dependent on factors external to the individual [2,6]. Specifically, the external HLOC can be addressed both to destiny/luck, and in this case, we speak of “Chance HLOC” when it is believed that fate plays a predominant role in determining personal well-being. When this responsibility is addressed to external figures, such as health workers, family and God, this is defined as “Powerful Others HLOC” [8]. 

The belief system about factors that govern the state of health is shaped by the surrounding environment, particularly by the family context and by the health behaviours and habits of the people around us [6].

Several studies have been conducted to examine the relationship between HLOC and people’s health behaviors [9,10,11]. Most of them agree that there is a positive association between internal HLOC and health choices. In fact, as the place of internal control increases, people’s predisposition towards healthy lifestyles increases, with an increase in physical activity and the adoption of a balanced diet [9,10]. A place of health control aimed at factors internal to the individual is also associated with better therapeutic compliance, as demonstrated by the work of Burkhart et al. [11].

The negative association between Chance HLOC and the adoption of healthy behaviors constantly emerges from several studies in literature [12,13]. Individuals who believe that fate strongly affects their state of health are more likely to consume alcohol, foods with high-fat content, to take less fiber and fruit in their diet and to exercise less physical activity [9,10], as well as to have less care oral hygiene and make less use of preventive health services [12,13]. Thus, it could be speculated that individuals who consider destiny as predictive of their state of health, are often not aware of their potential in terms of health and how much it can affect their psychophysical well-being. They do not feel directly responsible for their health condition and health and, therefore, may be more prone to risky behaviours [6].

Given the fundamental role of parents in providing for the psychophysical well-being of the child, especially in the first years of life, Devellis et al. has underlined the importance of investigating the HLOC of parents through the use of a specific scale, targeted for this category [14].

Given the importance of PHLOC in protecting and promoting infants’ health, we conducted a study aimed at identifying the Health Locus of Control of parents and to evaluate whether there are any correlations between PHLOC and socio-demographic characteristics of both parents and infants.

## 2. Materials and Methods

We conducted a single-centre cross-sectional observational study in a third-level clinical setting at the Fondazione IRCCS Ca’ Granda Ospedale Maggiore Policlinico of Milan, Italy. The population was enrolled at the outpatient clinics of the follow-up service from October 2019 to February 2020.

We included parents of term and preterm infants, at the first or second neonatal follow-up visit after hospital discharge (between 4 and 10 days of neonatal age) and/or undergoing neonatal screening at 15 +/− 1 day of neonatal age. The only exclusion criteria was an insufficient comprehension of the Italian language. Enrolment took place after the parents’ written informed consent. The present study was approved by the Ethics Committee of Fondazione IRCCS Ca’ Granda Ospedale Maggiore Policlinico (17 July 2018) with approval reference number 578_2018. Written informed consent was obtained from all mothers enrolled and from both parents.

At the enrolment, parents were given the Parent Health Locus of Control Scale (PHLOC scale), in the Italian version, validated in the study by Bonichini et al. This instrument consists of 28 items that assess parents’ beliefs about variants that may affect their child’s health [8]. The items identify six categories of subjects that parents believe could be responsible for their child’s health:
Parents: how responsible parents feel for their child’s health (7 items);Health professionals, i.e., the role of doctors and nurses in ensuring the child’s psychological and physical well-being (5 items);Mass media: the influence of television and magazines on children’s health (3 items);Divine/God: the importance that parents attach to the influence of the Divine on their child’s health (3 items);Fate: whether parents believe that their child’s well-being is primarily a matter of good/bad luck (5 items);The child: the influence the child has on his/her own health (5 items).

The categories ‘Parent’ and ‘Child’ (1 and 6) refer to an internal HLOC, whereas health professionals, the media, God and fate (2; 3; 4; 5) identify an external HLOC. The external HLOC is in turn divided into “Powerful Others” HLOC, represented by health professionals and God, and “Chance” HLOC, represented by fate.

For each item, parents have to express their degree of approval or disapproval through a 6-point Likert scale where score 1 equals to completely disagree and score 6 to completely agree with the assertion under investigation.

In addition, enrolled parents were requested to fill a questionnaire investigating the following sociodemographic variables of the parent (sex, age, marital status, level of education, work activity and if carried out in the health sector, citizenship of both parents and presence of other children), and of the newborn: sex, days of life, gestational age at birth in weeks and ward of admission.

The PHLOC scale and the questions investigating the socio-demographic variables were distributed to only one parent per couple (the only parent present at the visit or the parent who had given his or her availability to fill the questionnaire).

Data were analysed through different descriptive analysis according to the variable’s characteristics (for qualitative variables: absolute frequencies and percentages; for quantitative variables: mean and standard deviation). Pearson’s chi-squared test was used to evaluate possible associations between the items collected with the questionnaire and the parental health locus of control (PHLOC scale). The statistical significance value was set at *p* < 0.05.

To investigate possible correlations between socio-demographic variables and the items of the PHLOC scale, the response frequencies were categorized dichotomously as follows: parental age was divided according to the median value of the age of the enrolled population (36 years); marital status was divided between partnered/married and all other conditions in which the mother was a single parent. The cut-off for educational level was set at 13 years of schooling (Bachelor’s degree or above). Gestational age (GA) was subdivided into full-term births (births ≥ 37 weeks GA) and preterm births (births < 37 weeks GA), while the clinical course of the newborn was identified on the basis of the department in which the newborn was admitted at birth: Well-Baby Nursery (late preterm or term infants with regular clinical course) or Neonatal Intensive or Intermediate Care Unit (preterm infants or any complication in the clinical course). The presence of any siblings was divided between no siblings and one or more siblings.

The answers of each item of the PHLOC scale were classified as “disagree with the assertion under consideration” (Likert scale score ≤ 3) and “agree with the assertion under examination” (Likert scale score > 3).

## 3. Results

During the data collection period at the neonatal follow-up clinics, 445 parents were screened for enrolment. Of these, 75 were excluded from the study for an insufficient comprehension of the Italian language. The final sample consisted of 370 parents. The socio-demographic characteristics of the sample are shown in Table 1.

### 3.1. Parent’s Characteristics

The sample consisted of 69% (N 255) female parents; the mean maternal age was 34.8 years, while the mean paternal age was 37.7 years. The large majority of respondents (92.3%) were Italian, while 7.7% had a different nationality.

The 88% declared to be married/cohabiting. Most of the parents (65%), had a bachelor’s degree or a master’s degree as their highest qualification.

The majority (96%) of the parents declared to be employed. Fourteen percent of the parents enrolled declared to work in healthcare. More than half of the parents, 58%, stated that they had no other children, while 36% stated that they had at least another child. 

### 3.2. Child’s Characteristics

Newborns were 50.3% female and 49.7% male. There was a total of 3 twin pairs in the sample. The mean age was 9.8 days, and the mean gestational age at birth was 38.3 weeks.

Most of the newborns (87%) had a physiogical clinical course and were admitted to the Well-Baby nursery, 4% were admitted to the Intermediate Care Unit and 3% to the Neonatal Intensive Care Unit. 

### 3.3. Parent’s Health Locus of Control (PHLOC) Results

The graph shown in Figure 1 shows the response percentages in the individual categories of the PHLOC scale, divided according to the score assigned on a 6-point Likert scale: score ≤ 3 “disagree with the assertion under examination”; score > 3 “agree with the assertion under examination”.

The highest percentages of responses with a score > 3, showing a high agreement of the parents enrolled, are found in the categories of “Parent” (90.5% of responses) and “Health professionals“ (77.4% of responses); the lowest percentages are instead found in the categories of “Divine” and “Destiny”, respectively, with 21.4% and 16.3% of responses with a score > 3. Regarding the answers with a score ≤ 3 on the Likert scale, the highest percentages were observed in the categories “Fate” (83.7%) and “Divine” (78.6%), while the lowest percentages were observed in the subgroups “Parent” (9.5%) and “Health workers” (22.6%).

The sample recorded a predominantly internal HLOC, with a mean score in the influence category ‘Parent’ of 5, followed by the category ‘Health workers’ with 4.6 points (Powerful Others HLOC).

The average scores of the six macro-areas of the scale are shown in Figure 2.

The highest value is recorded in the influence category “parent”, with an average score of 5. The second highest score is recorded in the influence category “health professionals” with an average value of 4.6 points, followed by the subgroups “child” and “mass media” with both an average score of 2.8. The “God/divine” category obtained an average value of 2.2 points, slightly above the “fate” category, which scored the lowest, with an average of 2.1 points.

### 3.4. Correlations between PHLOC Scale Categories and Socio-Demographic Characteristics of the Sample

We performed a Pearson’s chi-squared test to evaluate possible correlations between PHLOC scale categories and socio-demographic characteristics of the enrolled population. We hereby present the results of the available answers or every PHLOC category, since not all enrolled parents have answered all items.

#### 3.4.1. Mass Media Category

Statistical analysis revealed significant correlations between the agreement to the “mass media” category (items 6–8–14) of the PHLOC scale the parent’s age, educational qualification and infant’s gestational age, as shown in Table 2.

The majority (66.8% N 127) of younger parents (aged ≤ 36 years) disagreed with item 6 “What my child sees in television advertisements may affect his or her health”, compared to 53.9% (N 76) of parents over 36 years of age (*p* = 0.017).

Item 6 of the “mass media” influence category was also significantly correlated with the gestational age of the newborn. Parents of term infants agreed more strongly with item 6 (40.1%, N 107) than parents of preterm infants (24.4%, N 11), with a significance level of 0.045. Regarding items 8 “Some of the comic books circulating nowadays might influence my child’s health” and 14 “What my child sees on TV might influence his or her well-being” were correlated with the parent’s educational qualification: 82.3% and 62.3% of parents with >13 years of education disagreed with item 8 and 14, respectively, compared to 69.3% and 48.9% of parents with ≤13 years of education.

#### 3.4.2. Health Professional Category

Items 10 and 20 of the macro area “Health workers” correlate significantly with gestational age of the newborn and number of children, respectively.

More than half (54.5%, N 146) of parents of term infants disagreed with item 10 “Only health professionals, such as doctors nurses or psychologists, can affect my child’s health” compared to 33.3% (N 15) of parents of preterm infants (*p* = 0.009).

First-time parents agreed more strongly (77.3%, N 150) with item 20 “Doctors have control over my child’s well-being” than their counterparts with more children (65%, N 89) at a significance level of 0.013 (Table 3).

#### 3.4.3. Parent Category

Regarding the ‘Parent’ category, a statistically significant correlation was found between item 2 and the variable ‘number of children’, with 95.4% (N 188) of parents with one child agreeing with item 2 ‘I have the ability to influence my child’s well-being’ compared to 88.7% (N 126) of parents with multiple children (*p* = 0.020), as shown in Table 4.

#### 3.4.4. Divine/God Category

The Item 11 belonging to the influence category “Divine” correlated significantly with the clinical course of the newborn. The majority of parents 81.9% (No. 222) of infants with an uncomplicated clinical course after birth disagreeing with item 11 “Only God can decide what will happen to my child’s health”, compared to 63.6% (No. 28) of parents of infants with a non-physiological course (*p* = 0.005) (Table 5).

#### 3.4.5. Fate Category

Regarding the category “Fate” of the PHLOC scale, item 18 (“Whether my child’s health will not deteriorate is just a matter of luck”) was significantly associated with the clinical course of the newborn (*p* = 0.007). Specifically, a higher percentage (89.6%, N 241) of parents of infants with a physiological course disagreed with item 18, compared to parents of infants with a non-physiological course (75%, N 33) (Table 6).

## 4. Discussion

The aim of this study was to investigate the Health Locus of Control of parents of infants born in third level clinical setting. Parents enrolled had a predominantly internal HLOC, followed by the category “Powerful Others HLOC”. Parents enrolled believe that they have a strong influence on the infant’s health with their behaviours and choices and, at the same time, they recognise the preponderant role of health workers in guaranteeing the baby’s well-being. On the contrary, parents of our sample consider God and fate to have little influence. The mass media and the child himself, on the other hand, play an intermediate role in the parents’ belief system. These results are in line with some studies present in the literature [15] in particular with the study by Bonichini et al. [8] in which 470 Italian mothers recorded the highest scores in the macro-areas “Parent” and “Health workers”, with an average of 4.9 and 3.55 points, respectively. As in the Bonichini et al. study [8], our samples consisted mainly of female parents (69%). This could have influenced the PHLOC subscales results, since locus of control has long been known to differ between males and females [16]. However, interestingly enough, our sample had overall a predominantly “internal LOC”, while female have been described to have a more “external LOC” than males [16].

In our study, a significant correlation emerged between the category “mass media” and the age of the parent, his/her educational qualification and the gestational age of the newborn. Younger parents (age ≤ 36 years) seem to consider mass media less influential on the health of the newborn compared to parents aged >36 years. It is well known that age is a factor that can influence HLOC. Previous studies have shown that an internal LOC is increasingly represented with increasing age and then gives way to a ‘powerful others’ LOC when in old age, the individual becomes aware of their mortality and must rely on others to maintain their good health [17]. In our work, intriguingly, younger parents give less importance to the influence of the mass media, particularly advertising, on their child’s health. This finding needs to be confirmed by future longitudinal studies exploring whether this remains the case throughout the child’s growth and childhood.

In addition, parents with an academic education believe less in the influence of mass media, compared to those with a non-academic education. Our result could be explained by the tendency of more educated people to believe that health depends on their own choices and behaviour and not on external factors [18]. This result overlaps with what has already been described in the literature: in fact, numerous studies have shown that a higher level of education is linked to a lower tendency to consider external factors, such as the mass media, as fundamental in influencing one’s health [19]. This seems to be linked to the belief that one’s abilities, exercised during higher education, are able to influence the positive or negative direction of one’s life.

Considering this difficult time of global health emergency due to the COVID-19 pandemic and rising rates of vaccine hesitancy, it is important to underline how the mass media can influence less educated parents and condition them in their choice of information and support from health professionals.

Parents of term infants (EG ≥ 37 weeks) seem to believe more in the influence of the media on their child’s health than parents of preterm infants, which may influence their health choices and lead them to devalue the role of health professionals. This result could mean that general health conditions (term vs. preterm) are an important factor in determining responses to the locus of control questionnaires. The different experience of the child’s hospitalisation could influence the parent’s conception of health and health determinants. Premature babies are a population considered at greater health risk, both in the short term but also concerning childhood and adulthood. It is already well known that premature birth places the child at a greater risk of developing non-communicable diseases, such as hypertension and metabolic syndrome. These particular pathologies can be delayed or reduced by preventive health behaviours regarding diet, physical activity and avoidance of negative conducts such as smoking.

During the period of hospitalisation health workers provide fundamental support to mothers of preterm infants, helping them to cope with the difficulties related to their child’s illness and fragile condition. Health workers are seen as those who “know”, both in terms of the services they provide and the knowledge they have about their baby [20,21]. In fact, when exploring the association with the “Health professionals” category, this study highlighted that parents of full-term infants believe less in the influence of health professionals on their child’s health than parents of preterm infants, whereas primiparity would appear to be related to a greater recognition of the importance of health workers in maintaining health.

This difference found could be explained by the different experience of hospitalisation between the parents of term and preterm babies; the latter, in fact, often have to cope with the transfer of their child to the Neonatal Intensive Care Unit (NICU), which determines an alteration in parenthood transition process. In this regard, the literature supports the existence of an association between a condition of chronic illness and/or clinical instability and a locus of health control directed at external factors, such as health professionals and fate [7,22]. This result would seem to be at odds with some studies showing an association between preterm birth and Chance HLOC. The study by Pichler-Stachl et al. [23], conducted in the first three postpartum days, recorded a significantly higher Chance HLOC score in mothers of infants with gestational age < 32 weeks than in mothers of term infants.

In our population, “health professionals category” was also associated with number of children. Specifically, first-time parents recognize more the role of health professionals for their babies health. In literature, there is mixed evidence on this association: previous studies confirm our results [7], while more recent ones find no significant association [8,24]. The study of Özcan et al. [24], conducted on a sample of 256 pregnant Turkish women, mostly unemployed and with a low to medium level of education and socioeconomic status, found no significant association between Powerful Others HLOC and the number of children. Our result could be explained by the greater self-efficacy perceived by parents with more than one child compared to first-time parents, and by the tendency of the latter to turn more often to external figures, such as health professionals [25].

Despite this result, our study suggests that first-time parents have a higher Internal HLOC than parents with more children (*p* = 0.020), believing that they are the main contributors to the baby’s well-being. This is in line with the study by Wallston et al. [7] in which first-time parents attending childbirth classes show a higher Internal HLOC than the other samples taken (healthy adults, university students and chronic patients). Our result could be explained by the greater sense of control over the birth event and the child’s health perceived by first-time parents in our setting, who regularly use preventive services, such as prenatal ultrasound check-ups and courses accompanying the birth, as already described by Wallston et al. [7].

Exploring the impact of the newborn’s clinical course, parents of children with a physiologically uncomplicated course are more reluctant to believe in the influence of God and fate on their child’s health. This finding is in line with other studies in literature, which supports the association between the experience of illness and a greater external locus of health control [22,23,26].

In agreement with previous studies [14,18], this work reaffirms the influence of sociocultural context and personal experience on HLOC orientation. Previous studies have highlighted how an “internal HLOC” and a “powerful others HLOC” are associated with greater sensitivity to preventive medicine, in terms of vaccination coverage [2,27], vitamin and iron intake during pregnancy [28] and dental caries prevention [12]. A “chance HLOC” is related to the adoption of health risk behaviours, such as smoking, increased consumption of alcohol and high-fat foods and refusal of one or more vaccinations [9,10,29]. In light of the influence between HLOC and socio-demographic variables, it is possible to identify those categories of parents who might be more prone to health risky behaviours and make less use of prevention services, such as being more hesitant towards paediatric vaccinations. According to the data obtained from our sample, older and less educated parents seem to believe more in the influence of the media on their child’s health, and this could expose them to dangerous behaviour, as they are not fully aware of their own health potential and can be influenced by media content, sometimes not supported by scientific bases [6]. On the contrary, first-time parents seem to have a higher internal HLOC, which is associated with greater use of preventive medicine services and better vaccination compliance [12,27].

The present study has a rather large and heterogeneous sample of parents, which made it possible to identify significant correlations between HLOC and the different subgroups of the sample. In addition, the self-administration procedure, and the guarantee of anonymous data processing made it possible to collect information that was not influenced by the presence of an interviewer (social desirability bias), enhancing the reliability of our data.

However, parents could have been influenced in their answers by the context of compilation. Although we enrolled term infants to explore differences between healthy babies’ parents’ HLOC with preterm infant’s PHLOC, the infant’s health overall and clinical course and not just prematurity could influence the parents HLOC. Moreover, our cross-sectional observational study made it possible to establish correlations between the different variables but not causal relationships, which are only possible with a longitudinal study. In fact, it must be considered that PHLOC could change over time as the parenthood experience develops. Finally, our study setting, although boasting a large attendance, may have influenced the characteristics of the sample, with some being more representative than others.

## 5. Conclusions

In conclusion, the assessment of the HLOC and of the socio-demographic variables that can influence it, is useful to identify those parents who are more prone to health-threatening behaviours, for the planning of targeted health education interventions, aiming at improving the overall health of the family and children and, at the same time, increasing trust in health professionals and authoritative sources of information. To achieve this ambitious goal, a targeted recall of parental history aimed at identifying risk factors could function as a screening at birth. A more in-depth study could also show the effectiveness of a personalized educational intervention based on these risk factors.

Further studies are needed to analyze how the COVID-19 pandemic has changed the PHLOC in different settings. In addition, a longitudinal study could evaluate how HLOC of parent’s could change in time during parenthood experience.

## Figures and Tables

**Figure 1 ijerph-19-05804-f001:**
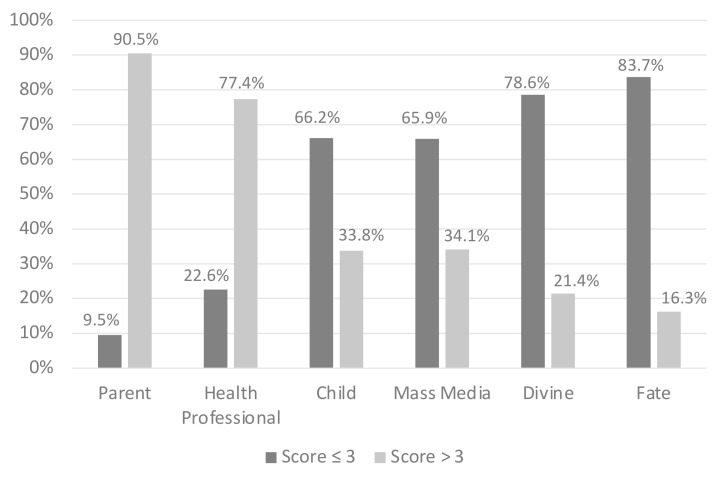
Parents’ disagreement (score ≤ 3) or agreement (score > 3) to the six subscales of PHLOC.

**Figure 2 ijerph-19-05804-f002:**
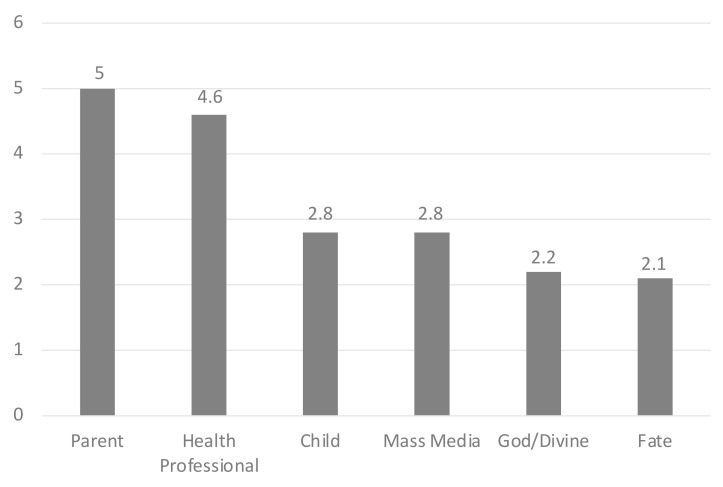
PHLOC sale categories and their mean Likert score.

**Table 1 ijerph-19-05804-t001:** Population’s characteristics. Variables are expressed as number of observations (%) or as mean ± standard deviation (SD).

Socio-Demographic Characteristics of the Sample	N	%
**Parent’s Sex**:		
Female	255	69%
Male	115	31%
**Marital Status**:		
Single	44	12
Married–Partnered	323	88
Divorced–Separated	3	1
**Educational Background**:		
Middle School	11	3
High school	100	27
Bachelor’s Degree	111	30
Master’s degree/Postgraduate	129	35
PhD or equal	19	5
**Employment**:		
Employed	355	96
Unemployed	15	4
**Works in healthcare**:		
Yes	52	14
No	318	86
**Newborn’s Sex**:		
Female	187	50
Male	185	50
**Prematurity**		
Full-term	320	86
Preterm	52	14
	Mean	SD
**Mother’s age (years)**	34.8	±4.8
**Father’s age (years)**	37.7	±6
**Newborn’s age (days)**	9.8	±9.7
**Newborn’s gestational age at birth (weeks)**	38.3	±2

**Table 2 ijerph-19-05804-t002:** Statistically significant correlations between “mass media” category and socio-demographic characteristics.

MASS-MEDIA				
Item 6. What My Child Sees in Television Advertisements May Affect His or Her Health Disagree (Score 1–3) Agree (Score 4–6)		Disagree (Score 1–3)	Agree (Score 4–6)	*p*-Value
Parent’s Age	≤36 years	**66.8% * (127)**	33.2% (63)	* 0.017
	>36 years	53.9% (76)	46.1% (65)	
Gestational Age	<37 weeks	75.6% (34)	24.4% (11)	* 0.045
	≥37 weeks	59.9% (160)	**40.1%* (107)**	
**Item 8. Some of the Comic Books Circulating Today Could Affect My Child’s Health**		**Disagree (Score 1–3)**	**Agree (Score 4–6)**	***p*-Value**
Educational Background	≤13 years	69.6% (64)	30.4% (28)	* 0.017
	>13 years	**82.3% * (191)**	17.7% (41)	
**Item 14. What My Child Sees on TV Can Influence His/Her Well-Being**		**Disagree (Score 1–3)**	**Agree (Score 4–6)**	***p*-Value**
Educational Background	≤13 years	48.9% (45)	51.1% (47)	* 0.026
	>13 years	**62.3% * (149)**	37.7% (90)	

**Table 3 ijerph-19-05804-t003:** Statistically significant correlations between “Health Professionals” category and socio-demographic characteristics.

Health Professionals				
Items 10. Only Health Professionals, Such as Doctors, Nurses or Psychologists, Can Have an Influence on My Child’s Health.		Disagree (Score 1–3)	Agree (Score 4–6)	*p*-Value
Gestational Age	<37 weeks	33.3% (15)	66.7% (30)	* 0.009
	≥37 weeks	**54.5% * (146)**	45.5% (122)	
**Items 20. Doctors Have Control over My Child’s Well-Being**		**Disagree (Score 1–3)**	**Agree (Score 4–6)**	***p*-Value**
Number of children	Only child	22.7% (44)	**77.3% * (150)**	* 0.013
	Other children	35% (48)	65% (89)	

**Table 4 ijerph-19-05804-t004:** Statistically significant correlations between “Parent” category and socio-demographic characteristics.

Parent Category				
Item 2. I Have the Ability to Influence My Child’s Well-Being		Disagree (Score 1–3)	Agree (Score 4–6)	*p*-Value
Number of children	Only child	4.6% (9)	**95.4% * (188)**	* 0.020
	Other children	11.3% (16)	88.7% (126)	

**Table 5 ijerph-19-05804-t005:** Statistically significant correlations between “Divine/God” category and socio-demographic characteristics.

God/Divine Category				
Item 11. Only God Can Decide What Will Happen to My Child’s Health		Disagree (Score 1–3)	Agree (Score 4–6)	*p*-Value
Newborn’s Clinical Course	physiological	**81.9%* (222)**	18.1% (49)	* 0.005
	non-physiological	63.6% (28)	36.4% (16)	

**Table 6 ijerph-19-05804-t006:** Statistically significant correlations between “Fate” category and socio-demographic characteristics.

Fate Category				
Item 18. Whether My Child’s Health Will Not Deteriorate Is Just a Matter of Luck		Disagree (Score 1–3)	Agree (Score 4–6)	*p*-Value
Newborn’s Clinical Course	physiological	**89.6% * (241)**	10.4% (28)	* 0.007
	non-physiological	75% (33)	25% (11)	

## Data Availability

The data presented in this study are available on request from the corresponding author. The data are not publicly available due to privacy issues.

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
