# Peer review of "Parent’s Health Locus of Control and Its Association with Parents and Infants Characteristics: An Observational Study"

_ijerph, 2022, doi:10.3390/ijerph19105804_

Round 1

Reviewer 1 Report

I really appreciate the opportunity to review this manuscript entitled “Parent’s Health Locus of Control and Its Association with Parents and Infants Characteristics: An Observational Study”. This is important to assess the locus of control and its association in families.  I only remark some issues (most of them in methods) in order to improve the quality of this manuscript.

The abstract is clear but it is important to explain the design, how many parents of full term and preterm infants were collected?

Introduction was well structure and shows the necessity for this research. In line 81 I think “the” is not necessary before studies. The aim of the paper is easy to understand.

At the methods section, there are some questions that should be review. If this study, is a part of a big research project (line 68) it should be explained.

About the inclusion criteria, why authors chose parents of full term and preterm only in the first days?  It is difficult to assess the influence of the parents in this period and may be it would have been interesting to do a follow-up. I also do not understand why parents were divided according to the median value of the age of the enrolled 148 population. Why variables were categorized dichotomously?

Results were clear. Discussion summarize and explain in a good way the finding but, from my point of view it would be interesting to discuss about gender differences (most participants are women). Which are the future lines of research?

Conclusions were correct.

Author Response

The abstract is clear but it is important to explain the design, how many parents of full term and preterm infants were collected?

We thank you the reviewer for the comment; according to your suggestion we have added in the abstract how many parents of full term and preterm infants were collected.

Introduction was well structure and shows the necessity for this research. In line 81 I think “the” is not necessary before studies. The aim of the paper is easy to understand.

We thank you the reviewer for the comment; we apologize for the type mistake that we have now corrected.

At the methods section, there are some questions that should be review. If this study, is a part of a big research project (line 68) it should be explained.

The study is not part of a big research project. In line 68 we refer to previous studies already available in the literature.

About the inclusion criteria, why authors chose parents of full term and preterm only in the first days?  It is difficult to assess the influence of the parents in this period and may be it would have been interesting to do a follow-up. I also do not understand why parents were divided according to the median value of the age of the enrolled 148 population. Why variables were categorized dichotomously?

We agree with the reviewer that enrolling parents later than in the first days after delivery would have been more interesting with regard to the aim of the study. However, in our Centre, term infants and preterm infants who have not developed major comorbidities undergo outpatient follow up visit within the first 15 days of life.

We decided to categorize the parents according to the median value of their age since this categorization allows for an adequate distribution of the enrolled subjects within the groups used for statistical analysis.

Results were clear. Discussion summarize and explain in a good way the finding but, from my point of view it would be interesting to discuss about gender differences (most participants are women). Which are the future lines of research?

We thank you the reviewer for the comment. According to your suggestion we have added a comment on the gender differences. (line 280)

We are planning to perform another study in order to evaluate whether the COVID 19 pandemic has affected the Health Locus of Control of parents.  We added the future lines of research in the conclusions.

Reviewer 2 Report

Thank you very much for the opportunity to review this interesting paper. Overall, the paper is well written, he research assumptions have been clearly presented, the research plan is simple and transparent as well as the presentation of the obtained results.

There are, however, some suggestions I would like to make:

  1. The article needs to be checked for minor errors, such as for example "from several the studies" or The We conducted".
  2. In the discussion of the results it is worth taking into account the factors that could have possibly influenced the answers given by the respondents. Baby's health, not just premature birth can affect the parents' responses. Moreover, health related locus of control can change over time due to many factors, such as normative and non-normative events, baby's health, parent's health, status of the relationship, overall well-being and so on. I would suggest to include the section limitation in the paper with some critical thoughts in it.
  3. I'm also concerned when it comes to results related to education level of a parent. The explanation in the paper seems to be oversimplified while the group was rather higher educated. When it comes to age of a parent, the results are not clearly explained, especially in the section concerning item 6, 8, and 14. Please, provide a deeper explanation. 
  4. I would suggest to repeat the study after six months or so (if it is possible to come back to respondents) and make it longitudinal. It would show the changes in HLOC over time and would be of course the material for another paper. 
  5. The explanation in what way the results 'can be useful to identify those parents who are more prone to health-threatening behaviours, for the planning of targeted health education interventions" would be appreciated and should be ad to the article. 

Author Response

The article needs to be checked for minor errors, such as for example "from several the studies" or The We conducted".

We thank you the reviewer for the comment; we have checked the article for minor errors.

In the discussion of the results it is worth taking into account the factors that could have possibly influenced the answers given by the respondents. Baby's health, not just premature birth can affect the parents' responses. Moreover, health related locus of control can change over time due to many factors, such as normative and non-normative events, baby's health, parent's health, status of the relationship, overall well-being and so on. I would suggest to include the section limitation in the paper with some critical thoughts in it.

We thank the reviewer for this important consideration. The aim of our study was to enroll term, physiological-course infants to explore the difference between their parent’s HLOC and that of preterm/non-physiological infants. We have implemented the limitations section of the study with these remarks (line 379) and have added a future line of a longitudinal study in the conclusions (line 393).

I'm also concerned when it comes to results related to education level of a parent. The explanation in the paper seems to be oversimplified while the group was rather higher educated. When it comes to age of a parent, the results are not clearly explained, especially in the section concerning item 6, 8, and 14. Please, provide a deeper explanation. 

We have implemented the discussion about the two aspects suggested by the reviewer: age result of item 6 (line 290) and education results of items 8 and 14 (line 301).

I would suggest to repeat the study after six months or so (if it is possible to come back to respondents) and make it longitudinal. It would show the changes in HLOC over time and would be of course the material for another paper. 

We thank the reviewer for this suggestion. We are planning to conduct a longitudinal study and have added the future research in the conclusions section.

The explanation in what way the results 'can be useful to identify those parents who are more prone to health-threatening behaviours, for the planning of targeted health education interventions" would be appreciated and should be ad to the article. 

We thank you the reviewer for the comment; we have added this consideration in the conclusion section